# Noninvasive and Convenient Screening of Metabolic Syndrome Using the Controlled Attenuation Parameter Technology: An Evaluation Based on Self-Paid Health Examination Participants

**DOI:** 10.3390/jcm8111775

**Published:** 2019-10-24

**Authors:** Yu-Jiun Lin, Chang-Hsien Lin, Sen-Te Wang, Shiyng-Yu Lin, Shy-Shin Chang

**Affiliations:** 1Department of Family Medicine, Taipei Medical University Hospital, Taipei 110, Taiwanpoanlinch@msn.com (C.-H.L.); wangader@yahoo.com.tw (S.-T.W.); daleslin@gmail.com (S.-Y.L.); 2Department of Family Medicine, School of Medicine, College of Medicine, Taipei Medical University, Taipei 110, Taiwan

**Keywords:** controlled attenuation parameter, FibroScan, metabolic syndrome, noninvasive, screening

## Abstract

Background: There is a medical need for an easy, fast, and non-invasive method for metabolic syndrome (MetS) screening. This study aimed to assess the ability of FibroScan to detect MetS, in participants who underwent a self-paid health examination. Methods: A retrospective cohort study was conducted on all adults who underwent a self-paid health examination comprising of an abdominal transient elastography inspection using FibroScan 502 Touch from March 2015 to February 2019. FibroScan can assess the level of liver fibrosis by using a liver stiffness score, and the level of liver steatosis by using the controlled attenuation parameter (CAP) score. The logistic regression analysis and receiver operating characteristic curve were applied to select significant predictors and assess their predictability. A final model that included all significant predictors that are found by univariate analysis, and a convenient model that excluded all invasive parameters were created. Results: Of 1983 participants, 13.6% had a physical status that fulfilled MetS criteria. The results showed that the CAP score solely could achieve an area under the curve (AUC) of 0.79 (0.76–0.82) in predicting MetS, and the AUC can be improved to 0.88 (0.85–0.90) in the final model. An AUC of 0.85 (0.83–0.88) in predicting MetS was obtained in the convenient model, which includes only 4 parameters (CAP score, gender, age, and BMI). A panel of predictability indices (the ranges of sensitivity, specificity, positive and negative likelihood ratio: 0.78–0.89, 0.66–0.82, 2.64–4.47, and 0.17–0.26) concerning gender- and BMI-specific CAP cut-off values (range: 191.65–564.95) were presented for practical reference. Conclusions: Two prediction systems were proposed for identifying individuals with a physical status that fulfilled the MetS criteria, and a panel of predictability indices was presented for practical reference. Both systems had moderate predictive performance. The findings suggested that FibroScan evaluation is appropriate as a first-line MetS screening; however, the variation in prediction performance of such systems among groups with varying metabolic derangements warrants further studies in the future.

## 1. Introduction

Metabolic syndrome (MetS) is a cluster of disorders, including insulin resistance/hyperglycemia, visceral adiposity [identified by large waistline (WL) or being overweight], atherogenic dyslipidemia (e.g., raised triglycerides (TG) or reduced high-density lipoprotein (HDL)), and endothelial dysfunction (characterized by elevated blood pressures) [1]. MetS significantly influences the development and deterioration of numerous diseases and is a crucial predictor of cardiovascular diseases [2,3,4]. Fortunately, numerous modifiable risk factors and corresponding practical intervention strategies regarding MetS were proposed [5,6,7], and hence, it is vital to identify the high-risk individuals nowadays to prevent the incidence and deterioration of metabolic derangement and its associated diseases. Nevertheless, the conventional procedures for the confirmation of MetS have depended on blood assays, which largely lead to a resistive mentality toward the procedures and are therefore nonconducive to long-term follow-up. Therefore, there is a high probability of the MetS population being under-identified, especially those without an obese body shape [8]. Consequently, there is a medical need for an easy, fast, and non-invasive method for MetS screening.

FibroScan is an ultrasound-based device equipped with the naval patented technology—Vibration-Controlled Transient Elastography (VCTE^TM^)—that was originally developed to assess the level of liver fibrosis and cirrhosis by using a liver stiffness score (the E score). Based on the transformation function of media stiffness and wave transmission velocity, the E score is calculated using a series of elastograms, which are simulated graphics of mathematical function of the time and depth of at least 10 shots of shear waves propagating through liver parenchyma transmitted using the probes mounted on the device. The multiple-shot shear wave and elastogram transformation are implemented with controlled vibration, energy, and algorithms on the identical volume of liver tissue to ensure measurement reliability [9,10,11,12]. The controlled attenuation parameter (CAP) score, denoting the liver steatosis, was embedded in instruments marketed after 2013. The score is derived from the amplitude attenuations of ultrasonic waves detected simultaneously from the E score measurement [12]. Owing to its merits of being noninvasive, safe, rapid (approximately 5–10 min for application), and the high-reproducibility of its automatically quantitative assessment outputs among operators [13,14,15], numerous studies have evaluated the performance of FibroScan in detecting chronic hepatitis [16,17] and various other liver disorders [12,18,19,20]. In addition to bedside inspection in hospitalized patients, the virtue of its portability was also manifested in population outreach screening [21]. In a street-based screening for hepatitis C virus (HCV) infection in drug users [22], the maneuver was excellently accepted by the target population. Various studies have reported the liver steatosis grading cut-off points of CAP score in patients with different liver disorders with satisfactory accuracy [12,23,24,25]. Overall, FibroScan has a multitude of merits, such as being noninvasive, safe, fast, convenient, portable, and highly accurate in the assessment of liver stiffness and steatosis.

A previous study found that the grades based on abdominal ultrasonography gave accurate information regarding liver steatosis and MetS in nonalcoholic, healthy people [26]. Moreover, considering the frequent coexistence and common pathological mechanisms of MetS and hepatopathy, such as steatohepatitis, liver inflammation, or the nonalcoholic fatty liver disease (NAFLD), it was proposed that liver steatosis, especially NAFLD, could be a hepatic manifestation of MetS [27,28,29]. Therefore, the FibroScan is investigated to be an option for the frontline screening of MetS in participants of a self-paid health examination in Taiwan. To the best of our knowledge, this is the first study to explore the utility of FibroScan in identifying people from the general population who have physical status and disorders that fulfill the MetS criteria. A subsequent objective was to explore its feasibility and reliability in screening for MetS without performing any simultaneous invasive procedures, such as blood assays.

## 2. Materials and Methods

### 2.1. Study Design

This was a single center, retrospective cohort study to assess the ability of FibroScan to detect MetS in participants of a self-paid health examination at the Health Management Center (HMC) of Taipei Medical University Hospital (TMUH). Data from March 2015 to February 2019 were analyzed.

### 2.2. Setting

The study was conducted at TMUH, and electronic records were also reviewed from the TMUH. TMUH is a private, tertiary care, 800-bed teaching hospital in Taiwan. The HMC of TMUH received approximately 60–70 visits each month. The study was approved by Investigational Review Board of the Taipei Medical University Hospital prior to the initiation of data collection.

### 2.3. Population and Data

The study included all Taiwanese adults, age > 18 years, for whom underwent a self-paid health examination comprising of an abdominal transient elastography inspection using FibroScan 502 Touch (Echosens, Paris, France). Previous genetic study suggested that 99.4% of Taiwanese can be classified as Han Chinese [30]. Participants were excluded if they took FibroScan for obvious symptoms or on the physician’s orders.

All the participants undertook the regular processes of the HMC. The participants were interviewed by well-trained personnel, who will verify the correctness of the participants’ self-completed questionnaire on demographics, existing medical conditions, and use of medications. In addition, the personnel will confirm adherence to the health examination prerequisites (e.g., overnight fasting for at least 8 hours) for the package chosen by the participant. Those who were found not to have followed the necessary prerequisites were suggested to book another appointment. Then, anthropometrics (weight, height, waist circumference, and arterial pressures) were measured. The instruments were regularly calibrated per the manufacturer’s specifications. The samples of blood, urine, and the specimen required per the chosen package were collected for laboratory tests. Regular laboratory test items included hemoglobin A1c (HbA1c), serum glutamic oxaloacetic transaminase (GOT), serum glutamic pyruvic transaminase (GPT), uric acid (UA), creatinine, blood urine nitrogen (BUN), red blood cell count (RBC), hemoglobin (Hb), hematocrit (Hct), MCH (mean corpuscular hemoglobin), MCV (mean corpuscular volume), mean corpuscular hemoglobin concentration (MCHC), platelet count, white blood cell count (WBC), percentage of neutrophils (Neu), lymphocytes (Lym), monocytes (Mono), eosinophils (Eso), and basophils (Baso), total protein, albumin, globulin, albumin/globulin ratio (A/G), total bilirubin, direct bilirubin, alkaline phosphatase (ALP), γ-glutamyl transpeptidase (γ-GT), total cholesterol, low-density lipoprotein cholesterol (LDL), HDL cholesterol, LDL/HDL ratio, total cholesterol/HDL ratio, TG, and fasting blood sugar (FBS). The glomerular filtration rate (GFR) was estimated using a simplified modification of diet in renal disease (MDRD) calculated based on creatinine, age, and gender using the IDMS-traceable MDRD formula [31] with an ethnicity-specific multiplier (0.945) reported by the Health Promotion Administration of Taiwan. For each FibroScan inspection, two scores were reported: controlled attenuation parameter (CAP score) and liver stiffness parameter (E score). The dashboard of a FibroScan manifests a CAP score only when an E score derived from identical signals is validated as successfully computed. The higher the E score, the higher is the transmission velocity and the level of liver stiffness. Furthermore, the higher the CAP score, the faster the wave amplitude attenuates, and the higher the level of liver steatosis. Notably, the adoption of probe sizes (M or XL) was based on the recommendation of the instrumental autodetection function.

### 2.4. Outcome

MetS was identified upon fulfillment of National Cholesterol Education Program Adult Treatment Panel III definition of MetS consensus of at least three out of the five symptoms [32]: large WL (≥80 cm for women and ≥90 cm for men), high TG (≥150 mg/dL) or use of medication to control TG, reduced HDL levels (<50 mg/dL for women and <40 mg/dL for men) or use of medication to control HDL, elevated blood pressure (BP; systolic BP ≥130 mmHg or diastolic BP ≥85 mmHg) or use of related medication to control blood pressure, and increased FBS (≥100 mg/dL) or use of related medication to control blood sugar. The classification of cut-off points was adopted from the National Cholesterol Education Program Adult Treatment Panel III definition consensus with ethnic-specific cut-off points for waist circumference [33] and an equality principle on the five disorders. 

### 2.5. Statistical Analysis

In Table 1, the baseline characteristics of the enrollees were described and compared among MetS cases and non-MetS participants. Categorical variables were presented as a frequency and percentage and were compared between MetS cases and non-MetS participants using the χ^2^ test. Continuous variables were presented as medians and interquartile range (IQR), and were compared between MetS cases and non-MetS participants using the Mann–Whitney U test.

A univariate logistic regression analysis was conducted using all the baseline characteristics variables, to identify predictor for MetS. The risk for MetS was expressed as an odds ratio (95% confidence interval), and shown in Table 2. 

A multivariable logistic regression model was first built by including all statistically significant variables in Table 2, but excluding variables with a direct contribution to MetS classification, which included HDL, ratios with components of HDL, WL, SBP, DBP, TG, and FBS. As a second step, the variables were removed sequentially if found to be not significant. However, the variable “Gender” was allowed to remain in the final model even though the *p*-value was not significant, as “Gender” was shown to be an important predictor for MetS by prior research. The components of the final model, and the risk for MetS can be found in Table 3. We also created another convenient model by removing all variables that required invasive procedures to determine, which included HbA1c, GPT, GFR, hemoglobin, and neutrophils. 

The predictabilities of both models were assessed accordant with receiver operating characteristic (ROC) curve, area under the curve (AUC), and relevant parameters [sensitivity, specificity, positive predictive value (PPV), negative predictive value (NPV), positive likelihood ratio (PLr), and negative likelihood ratio (NLR)]. The ROC curves can be found in Figure 1, and the Optimal Youden index-based cut-off points and the associated screening performance index can be found in Table 4. 

Finally, internal calibration using decile calibration plots for a graphic diagnosis on the goodness-of-fit, can be found in Figure 2. The Hosmer–Lemeshow test was also applied to evaluate the goodness-of-fit of candidate models. All the statistical analyses were performed using SAS 9.4 (SAS Inc., Cary, NC, USA). A *p*-value < 0.05 was considered statistically significant for all analyses.

### 2.6. Ethics

The Investigational Review Board of the Taipei Medical University Hospital approved this study (TMUH; TMU-JIRB No.: N201903080), in accordance with the original and amended Declaration of Helsinki. The necessity of written informed consent was waived owing to the retrospective chart-reviewing design of data collection and a series of appropriate maneuvers: identity code confidentiality, de-identification, data protection, and study result reporting restrictions.

## 3. Results

Overall, 1983 participants were included, with a mean age of 44.9 years, and ± standard deviation of 11.8 years. For the entire cohort, men accounted for 46.8%, and the majority of the participants were normal-BMI range somatotype (49.2%). The distribution of the other somatotypes using BMI cut-off values of <18.5 for underweight, ≥24 for overweigh, and ≥27 for obese are as follows: underweight 4.4%; overweight 26.5%; and obese 19.9%. For the entire cohort, the prevalence of MetS is 13.6%. Therefore, the values for the MetS related variables tended to be normal, when the mean± standard deviation values are being investigated. The mean SBP was 117.8 ± 16.7 mmHg, the mean DBP was 74.4 ± 10.8 mmHg, the mean total cholesterol was 189.8 ± 34.9 mg/dL, mean HDL was 55.2 ± 15.9 mg/dL, mean LDL was 124.5 ± 32.8 mg/dL, and the mean fasting blood glucose was 94 ± 20.1 mg/dL. 

Table 1 compares the baseline clinical characteristics of 269 MetS cases and 1714 non-MetS participants.

Table 2 displays the results of the univariate logistic regression analysis for predicting MetS status. Measurements that exhibited significant positive association with MetS status were E score, CAP score, age, male gender, BMI, SBP, DBP, HbA1c, GOT, GPT, UA, GFR, creatinine, RBC, Hb, Hct, MCHC, WBC, segmented neutrophils, alkaline phosphatase, γ-GT, LDL/HDL ratio, total cholesterol/HDL ratio, TG, and fasting glucose. Measurements that exhibited significant negative association were Lym, and HDL. 

Table 3 lists the components of the two models (convenient versus final), and the associated risks of MetS status. In general, similar associated risks of MetS were observed for the two models. The only exception is gender, where the risk changed from 0.86 (0.53–1.39) in the final model to 1.57 (1.14–2.15) in the convenient model, when the variables that require invasive blood draw were removed. 

The AUC is a global accuracy index that combines both sensitivity and specificity. In terms of AUC, the final model has significantly better performance than the convenient models, but the convenient model still exhibited reasonably good predictability (AUC: 0.88 vs. 0.85, *p* < 0.0001).

Figure 1 compares the summary ROC curves of the convenient model, final model, age, CAP, and BMI. Based on AUC, the final model has the highest accuracy in predicting MetS status (0.878, CI, 0.854–0.901), followed by the convenient model, BMI, and CAP respectively. CAP by itself has a reasonable AUC of 0.789 (CI:0.757–0.82), and has better accuracy than age, hemoglobin A1c, GPT, or GFR. 

Table 4 shows the optimal Youden index-based cut-off points of CAP of the convenient model and final model. In general, the convenient model has higher sensitivity than the final model (0.89 vs. 0.78), but lower specificity than the final model (0.66 vs. 0.82).

Figure 2 shows the decile calibration plots for screening the presence of metabolic syndrome. In general, both models showed a fair agreement between the observed probabilities of being MetS and the predicted ones, and all the 95% CI of observed probabilities covered the mean predicted probabilities of every decile. In addition, the results of the Hosmer–Lemeshow goodness-of-fit test indicated that both fitted models were adequately fitted (*p* = 0.4190 and 0.3563 for final and convenient models, respectively).

## 4. Discussion

This study assessed the ability of FibroScan measurements to detect MetS statues. FibroScan can assess the level of liver fibrosis by using a liver stiffness score, and the level of liver steatosis by the CAP via a non-invasive method. The results showed that the CAP score solely could achieve an accuracy of 79%, and the accuracy can be improved to 88% in the final model that includes other parameters that required a blood draw. An accuracy of 85% in predicting MetS was also obtained in the convenient model, which includes only 4 parameters (CAP score, gender, age, and BMI).

The ascertainment of MetS has relied on blood assays [1]. The typical repulsive mentality toward the invasiveness and painfulness of phlebotomy hinders population-based case identification. Several researchers have advocated regarding finding alternative first-line strategies to sieve out never-detected cases of MetS through economical and labor-dependent ways (e.g., anthropometrics) [34,35,36] or by using noninvasive but radioactive instruments (e.g., bioelectrical impedance analysis or dual-energy X-ray absorptiometry) [36,37,38]. By contrast, FibroScan screening is devoid of any invasive and radioactive maneuvers apart from having virtues of simple implementation, rapid measurement, speedy results, easy -portability, and acceptable accuracy, and provides details regarding multi-facet indices (liver stiffness and steatosis). Moreover, the CAP score, specially designed for measuring visceral adiposity, has manifested as having a stronger relationship to MetS than any other adiposity indices in postmenopausal women [36]. Despite being an essential risk factor of MetS development, interventions for obesity were predicted to have the most significant reduction in 10-year MetS prevalence [39]. Hence, long-term surveillance of visceral adiposity should be a crucial maneuver for MetS prevention, and the results here presented feasible routes.

To the best of our knowledge, this is the first study to investigate the utility of FibroScan in screening community-based population who fulfill the MetS criteria. Although there is no report on the use of FibroScan in a community-based population, the CAP cut-off that is found for the convenient model in this study was consistent with a previous study on severe steatosis patients. We found a CAP cut-off of 292 for identifying Taiwanese female participants with MetS, and a French study found a CAP cut-off 296 for identifying severe steatosis participants with chronic liver disease [23]. Another French study also reported that a CAP cut of 282 could identify severe steatosis patients with chronic hepatitis B [24].

Nonetheless, there are several important limitations to this study. Firstly, it has been reported that the detection failure of FibroScan (both E and CAP score) could be varied owing to specific thoracic anatomy structures and fat mass repartition, which is heterogeneous depending on age, gender, ethnicity, and metabolic disorder conditions [12,40]. For example, characteristics of high-riding liver, hyperinflated lungs, ascites, or thick subcutaneous fat in the trunk could reduce the success rate of measurement. Although evidence supported that the measurement success rate does not alter the accuracy of chronic liver disorder detection [12,41], there is still a need to clarify the characteristics associated with high measurement failure for the target population and establishing standards of procedures (e.g., rules for picking probes of appropriate sizes) to ensure a stable reliability before regularly applying this instrument. Notably, we observed very few measurement failures in this cohort of participants who underwent a self-paid health examination. Secondly, this study investigated only health-conscious participants that underwent a self-paid health examination. Participants with malignancy, end-stage chronic conditions, complex medical conditions, bedridden status, or poor socioeconomic status are unlikely to be enrolled in our cohort. Therefore, a lower prevalence of MetS was reported for this study (13.6%) as compared to the prevalence reported by an epidemiology study using the general population (13.6–25.5%) [42]. It is likely that the reliability parameters, such as sensitivity and specificity that were estimated, would be the reference level of lower limits when the FibroScan is tested at a population-based setting.

Thirdly, this study concerned mainly Han Chinese population residing in Taiwan, and the study results may not apply to subjects with other ethnic backgrounds. Given that metabolic disorders were reported to have ethnic differentiation [28], this study needs to be repeated and validated in other populations, to confirm that FibroScan evaluation is indeed appropriate as a tool for first-line MetS screening. 

## 5. Conclusions

To date, MetS is diagnosed mainly by blood assays. Our study, which used a large cohort of participants to derive predictors of MetS and validate models of Mets, found that fatty liver may be associated with Mets, and four non-invasive parameters (the CAP score, together with age, gender, and BMI) can be used to screen for MetS participants with reasonable accuracy. The convenient model may be a valuable tool for screening MetS, in subjects who were resistant to the inconvenience of overnight starvation or the painful blood assays. However, considering the retrospective nature of this study, a sufficiently powered prospective cohort study is needed to conclusively address the usefulness of the convenient model as a first-line MetS screening tool.

## Figures and Tables

**Figure 1 jcm-08-01775-f001:**
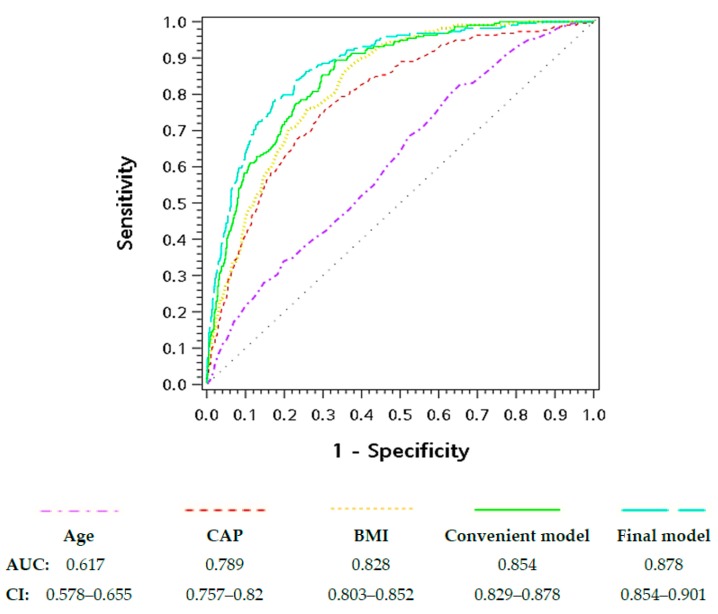
Receiver operating characteristic curves for predicting the presence of metabolic syndrome. Note: The AUC results not shown for the clarity of illustrating were hemoglobin A1c: 0.736 (CI, 0.698–0.775), GPT: 0.709 (CI, 0.672–0.746), and GFR: 0.64 (CI, 0.601–0.679).

**Figure 2 jcm-08-01775-f002:**
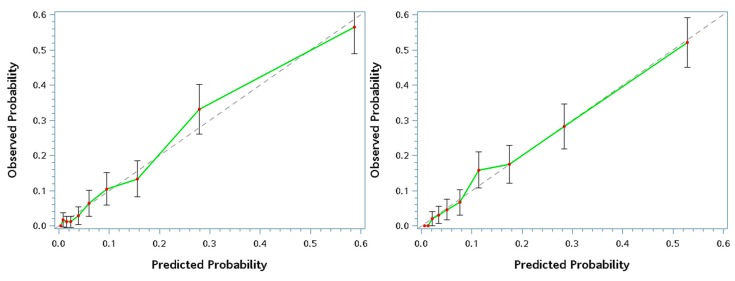
Decile calibration plots for screening the presence of metabolic syndrome. Left panel plots the convenient model, and the right panel plots the final model. The discontinuous line is the predicted probability, while the solid green line is the observed probability.

**Table 1 jcm-08-01775-t001:** Summary of the study sample characteristics by metabolic syndrome status (*n* = 1983).

Variable	Metabolic Syndrome		*p*-Value
Median (IQR)/Count (%)	No (*n* = 1714)	Yes (*n* = 269)	
FibroScan measurements			
E score (kPa)	4.2 (3.5–4.9)	5 (4.3–5.9)	<0.0001 *
CAP score (dB/m)	236 (207–272)	300 (264–333)	<0.0001 *
Demographics and anthropometrics			
Age, years	44 (36–52)	47 (41–57)	<0.0001 *
Gender			
Female	958 (55.9)	97 (36.1)	<0.0001 *
Male	756 (44.1)	172 (63.9)	
BMI (kg/m^2^)	23.1 (21–25.5)	27.4 (25.2–30)	<0.0001 *
Systolic blood pressure (mmHg)	113 (104–124)	135 (124–142)	<0.0001 *
Diastolic blood pressure (mmHg)	72 (66–79)	86 (78–91)	<0.0001 *
Comorbidity-related index			
Hemoglobin A1c (%)	5.3 (5.1–5.5)	5.7 (5.4–6.1)	<0.0001 *
GOT (U/L)	20 (17–24)	23 (19–31)	<0.0001 *
GPT (U/L)	18 (13–26)	28 (19–44)	<0.0001 *
Uric acid (mg/dL)	5.2 (4.3–6.4)	6.3 (5.4–7.2)	<0.0001 *
GFR (ml/min/1.73 m^2^)	97.03 (82.1–112.76)	86.23 (72.53–99.24)	<0.0001 *
Creatinine (mg/dL)	0.7 (0.6–0.9)	0.9 (0.7–1)	<0.0001 *
Blood urine nitrogen (mg/dL)	12 (10–15)	13 (11–16)	0.0005 *
HBsAg (IU/mL)			
Negative	1401 (81.7)	231 (85.9)	0.51
Positive	189 (11)	27 (10)	
Anti-HBs (mIU/mL)			
Negative	543 (31.7)	85 (31.6)	0.731
Positive	1040 (60.7)	171 (63.6)	
Anti-HCV (S/CO)			
Negative	1480 (86.3)	228 (84.8)	0.8731
Positive	93 (5.4)	15 (5.6)	
Biochemical index			
Red blood cell count (10^6^/uL)	4.68 (4.37–5.05)	4.96 (4.63–5.23)	<0.0001 *
Hemoglobin (g/dL)	13.9 (13–15)	14.8 (13.8–15.6)	<0.0001 *
Hematocrit (%)	41.3 (38.7–44.4)	43.7 (41.2–45.9)	<0.0001 *
MCH (pg)	30.1 (28.8–31.1)	30.1 (28.9–31.1)	0.5572
MCV (fL)	89.2 (85.9–91.8)	88.6 (85.3–91.1)	0.0662
MCHC (g/dL)	33.6 (33.1–34.2)	33.9 (33.4–34.4)	<0.0001 *
Platelet count (10^3^/uL)	233 (197–271)	234 (202–271)	0.8476
White blood cell count (10^3^/uL)	5.88 (4.87–7.15)	6.76 (5.81–8.05)	<0.0001 *
Neu (%)	57.55 (52.5–62.9)	59.1 (54.4–64.6)	0.0103 *
Lym (%)	30.8 (26.1–35.7)	30.2 (25.4–34.7)	0.0538
Mono (%)	7.6 (6.5–8.7)	7.2 (6.2–8.8)	0.0411 *
Eso (%)	2.2 (1.4–3.5)	2.3 (1.5–3.4)	0.9791
Baso (%)	0.7 (0.5–0.9)	0.7 (0.5–0.9)	0.0231 *
Total protein (g/dL)	7.4 (7–7.7)	7.4 (7.1–7.7)	0.0577
Albumin (g/dL)	4.6 (4.4–4.8)	4.6 (4.4–4.8)	0.6006
Globulin (g/dL)	2.7 (2.5–3)	2.8 (2.5–3.1)	0.1019
Albumin/globulin ratio	1.7 (1.5–1.9)	1.7 (1.5–1.9)	0.2601
Total bilirubin (mg/dL)	0.6 (0.4–0.8)	0.6 (0.4–0.8)	0.4768
Direct bilirubin (mg/dL)	0.2 (0.2–0.3)	0.2 (0.2–0.3)	0.6328
Alkaline phosphatase (U/L)	60 (50–72)	66 (55–80.5)	<0.0001 *
γ-Glutamyl transpeptidase (U/L)	16 (12–25)	27 (20–43.5)	<0.0001 *
Total cholesterol (mg/dL)	187 (165–210)	194 (165–220)	0.0242 *
LDL (mg/dL)	121 (101–143)	133 (106–158)	<0.0001 *
HDL (mg/dL)	55 (47–66)	39 (35–45)	<0.0001 *
LDL/HDL ratio	2.2 (1.6–2.9)	3.4 (2.7–4.1)	<0.0001 *
Total cholesterol/HDL ratio	3.3 (2.7–4.2)	4.95 (4.1–5.8)	<0.0001 *
TG (mg/dL)	85 (62–118)	178 (143–233)	<0.0001 *
FBS (mg/dL)	90 (85–95)	102 (92–111)	<0.0001 *

Note: The descriptive statistics were mean (IQR) for continuous variables and count (percentage) for categorical variables. The abbreviations are listed below. IQR = interquartile range. E score = Elastic modulus measured by the transient elastography of FibroScan. CAP score = Controlled attenuation parameter measured using FibroScan. BMI = Body mass index. GOT = Aspartate aminotransferase/serum glutamic oxaloacetic transaminase. GPT = Alanine aminotransferase/serum glutamic pyruvic transaminase. GFR = Glomerular filtration rate (GFR) estimated using simplified modification of diet in renal disease (MDRD). HBsAg = The surface antigen of Hepatitis B virus (Negative: <0.05). Anti-HBs = Antibody to hepatitis B surface antigen (Negative: <10). Anti-HCV = Antibodies against hepatitis C virus (Negative: <1). MCH = Mean corpuscular hemoglobin. MCV = Mean corpuscular volume. MCHC = Mean corpuscular hemoglobin concentration. Neu = Neutrophils. Lym = Lymphocytes. Mono = Monocytes. Eso = Eosinophils. Baso = Basophils. LDL = Low-density lipoprotein cholesterol. HDL = High-density lipoprotein cholesterol. TG = Triglyceride. FBS = Fasting blood sugar. * = *p* < 0.05

**Table 2 jcm-08-01775-t002:** Univariate logistic regression analysis for the presence of metabolic syndrome.

Variables	Odds Ratio	95% CI		*p*-Value
FibroScan measurements				
E score (kPa)	1.02	1.02	1.02	<0.0001 *
CAP score (dB/m)	1.02	1.01	1.04	0.003 *
Demographics				
Age (Years)	1.03	1.02	1.04	<0.0001 *
Gender, male vs. female	1.59	1.17	2.15	0.003 *
Anthropometrics				
BMI (kg/m^2^)	1.28	1.22	1.34	<0.0001 *
Systolic blood pressure (mmHg)	1.1	1.08	1.12	<0.0001 *
Diastolic blood pressure (mmHg)	1.06	1.05	1.07	<0.0001 *
Comorbidity-related index				
Hemoglobin A1c (%)	1.6	1.38	1.86	<0.0001 *
GOT (U/L)	1.01	1	1.03	0.007 *
GPT (U/L)	1.01	1.01	1.02	0.0005 *
Uric acid (mg/dL)	1.2	1.09	1.34	0.0005 *
GFR (ml/min/1.73 m^2^)	0.99	0.98	1	0.005 *
Creatinine (mg/dL)	2.47	1.42	4.3	0.001 *
Blood urine nitrogen (mg/dL)	1.01	0.98	1.05	0.548
HBsAg, positive vs. negative	0.89	0.56	1.42	0.624
Anti-HBs, positive vs. negative	1.19	0.86	1.63	0.287
Anti-HCV, positive vs. negative	1.14	0.6	2.15	0.686
Biochemical index				
Red blood cell count (10^6^/uL)	1.43	1.05	1.94	0.022 *
Hemoglobin (g/dL)	1.34	1.16	1.55	<0.0001 *
Hematocrit (%)	1.09	1.03	1.15	0.001 *
MCH (pg)	1.02	0.96	1.08	0.551
MCV (fL)	1	0.98	1.02	0.74
MCHC (g/dL)	1.32	1.11	1.57	0.002 *
Platelet count (10^3^/uL)	1	1	1	0.979
White blood cell count (10^3^/uL)	1.08	1.01	1.15	0.027 *
Neu (%)	1.04	1.02	1.06	0.0002 *
Lym (%)	0.97	0.95	0.99	0.002 *
Mono (%)	0.95	0.88	1.03	0.231
Eso (%)	0.93	0.86	1.01	0.077
Baso (%)	0.66	0.42	1.03	0.066
Total protein (g/dL)	1.18	0.85	1.65	0.324
Albumin (g/dL)	0.67	0.38	1.19	0.173
Globulin (g/dL)	1.42	0.99	2.02	0.055
Albumin/globulin ratio	0.67	0.39	1.14	0.139
Total bilirubin (mg/dL)	0.99	0.67	1.47	0.972
Direct bilirubin (mg/dL)	1.02	0.57	1.81	0.954
Alkaline phosphatase (U/L)	1	1	1.01	0.036 *
γ-Glutamyl transpeptidase (U/L)	1.01	1	1.01	0.008 *
Total cholesterol (mg/dL)	1	1	1	0.901
LDL (mg/dL)	1	1	1.01	0.323
HDL (mg/dL)	0.89	0.88	0.91	<0.0001 *
LDL/HDL ratio	2.15	1.82	2.54	<0.0001 *
Total cholesterol/HDL ratio	2.02	1.76	2.32	<0.0001 *
Triglyceride (mg/dL)	1.02	1.01	1.02	<0.0001 *
Fasting glucose (mg/dL)	1.03	1.02	1.04	<0.0001 *

Note: The abbreviations are explained in footnote of Table 1.

**Table 3 jcm-08-01775-t003:** Multivariate logistic regression analysis for metabolic syndrome.

Models	Convenient Model				Final Model			
Variables	OR	95% CI		*p*-Value	OR	95% CI		*p*-Value
CAP score (dB/m)	1.01	1.01	1.02	<0.0001 *	1.01	1	1.01	<0.0001 *
Gender, male vs. female	1.57	1.14	2.15	0.0055 *	0.86	0.53	1.39	0.5346
Age (years)	1.04	1.03	1.05	<0.0001 *	1.03	1.01	1.05	0.0005 *
BMI (kg/m^2^)	1.28	1.22	1.34	<0.0001 *	1.28	1.21	1.36	<0.0001 *
Hemoglobin A1c (%)					1.57	1.35	1.82	<0.0001 *
GPT (U/L)					1.01	1.01	1.02	0.0016 *
GFR (ml/min/1.73 m^2^)					0.99	0.98	1	0.0233 *
Hemoglobin (g/dL)					1.25	1.07	1.46	0.0056 *
Neutrophils (%)					1.03	1	1.05	0.0217 *
AUC ^†^	0.85	0.83	0.88	<0.0001 *	0.88	0.85	0.9	-

†: A *p*-Value of less than 0.05 indicated the AUC (area under the curve) of the convenient model was significantly different from that of the final model. * = p < 0.05.

**Table 4 jcm-08-01775-t004:** Optimal Youden index-based cut-off points and the associated screening performance index by classification models.

Models	Prediction Performance Parameters							Gender- and BMI-Specific Cut-Off Points of CAP ^†^							
	Thd	Sen	Spe	PPV	NPV	PLr	NLr	(Male)				(Female)			
								*Whole*	(18.5)	(24)	(27)	*Whole*	(18.5)	(24)	(27)
Convenient	0.09	0.89	0.66	0.28	0.97	2.64	0.17	254.14	372.03	254.57	191.65	292.28	410.17	292.71	229.78
Final	0.15	0.78	0.82	0.39	0.96	4.47	0.26	390.84	564.95	391.48	298.54	371.79	545.9	372.42	279.49

Note: Thd = threshold. Sen = sensitivity. Spe = specificity. PPV = positive predictive value. NPV = negative predictive value. PLr = positive likelihood ratio. NLr = negative likelihood ratio. †: The cut-off points of CAP (dB/m) were calculated in scenarios that adopted the sample means of age (44.88 years), hemoglobin A1c (5.49%), serum glutamic pyruvic transaminase (24.49 U/L), glomerular filtration rate estimated using simplified modification of diet in renal disease (97.53 mL/min/1.73 m^2^), hemoglobin (14 g/dL), and neutrophils (57.66%) in the models with gender and BMI levels as specified in the column title labels. For scenarios with column titles labeled “*Whole*”, the overall mean BMI (24.02 kg/m^2^) was used in calculation; for those labeled by numbers (18.5, 24, or 27), the specific numbers were used.

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
