# Peer review of "Noninvasive and Convenient Screening of Metabolic Syndrome Using the Controlled Attenuation Parameter Technology: An Evaluation Based on Self-Paid Health Examination Participants"

_jcm, 2019, doi:10.3390/jcm8111775_

Round 1

Reviewer 1 Report

Authors have adequately replied to my questions and made the necessary changes to the MS.Thus, the paper is improved in this revised version.

Reviewer 2 Report

The authors have adequately addressed all my comments.

This manuscript is a resubmission of an earlier submission. The following is a list of the peer review reports and author responses from that submission.

Round 1

Reviewer 1 Report

The research aim herein described was to assess the utility of controlled attenuation parameter (CAP) as a non-invasive screening tool for metabolic syndrome (MetS)

This non-invasive diagnostic/screening approach is novel and if proved to be sufficiently accurate could enable early detection of patients with metabolic syndrome, thus at risk for cardiovascular disease and type 2 diabetes, for targeted intervention.

The authors conclude that Fibroscan evaluation is appropriate as first-line screening for MetS.

The research is overall well designed and properly described. However, there are some questions that need to be addressed.

1- The Introduction well written and provides the necessary information for the reader to understand the rationale of conducting this research. Nevertheless, the reference list needs extensive revision, as in many occasions is out of focus and excessive. This applies to line 43 (refs 2,3,4,5,6,7,8) when refs 2,6 and 7 would suffice and several do not apply. The same holds true for line 44 referencing quoting 15 refs to support the statement that MetS risk factors are modifiable by target interventions. And again in line 80, where 11 references are found to support the statement that NAFLD could be considered a manifestation of MetS, when 3 refs ( 44, 49, 50) would suffice and several others do not apply. The authors need to conduct a thoughtful review of the references used in order not to be misleading for the readership.

2-In the Methods section, the authors do not clearly state that the parameters used for MetS assessment were not complete, since the use of drugs for glucose, blood pressure and lipid parameters lowering, were not part of the protocol as it should have been in order to enable a correct identification of patients with the condition. Reasons for not having assessed drug use during the patient interview, needs to be provided.

3-In the Results section, line 165, based on the BMI cur-off values it can be assumed that this was a patient population from an Asian ethnic background. However there is no clear description in the results regarding the ethnic background of the population. This information should be included in the patient demographics.

4-The prevalence of MetS in this study was 15.6%, how does this figure compares to the general background population, is there any epidemiological data that the authors could provide in order to assess whether this percentage was according to the expectations.

5- Figure and tables insertion along the text is not the most adequate to facilitate the data appreciation. Figure 1 should be depicted immediately after line 226 and figure 4 at line 230 after “…illustrated”. Figure 2 caption should also be presented at the end of the paragraph ( line 234).

6-In the discussion section the authors state that in the Asian population CAP measurement failure is rare, but do not provide a reference for such an evidence.

7-As limitations the authors should acknowledge that this study results need to be repeated and validated in other patient populations, in particular including patients from other ethnic backgrounds, since the conclusions retrieve only apply to an Asian population.

Reviewer 2 Report

In this study, the Authors evaluated the predictability of liver steatosis and the attenuation of this parameter in patients with a predisposition for Metabolic syndrome (MetS). The Authors performed the study on 1983 patients from 2015 to 2019,  evaluating the routine blood analysis and comparing the results with the FibroScan method outcomes. FibroScan is an ultrasound instrument with patented  technology and that measures the levels of liver fibrosis and cirrhosis by using a liver stiffness score (E score).

Even if the study aim is of interest in the field, the novelty of project design should be clarified and the methodological analysis should be improved. The results section should be better developed and explained and the text should be deeply revised. Thus, in my opinion, the paper is not suitable for publication in the present form and the following are some issues the Authors should address to improve the quality of the manuscript.

Points:

The Material and Methods section is poor of details. The Authors should improve the description of the clinical tests used for the evaluation of MetS and FibroScan. The advantage in using Fibroscan should emerge and must to be better argued. Results section appears confusing and needs to be better described. Results section, the Authors should specify the meaning of the numbers in parentheses. Discussion section: The Authors should better discuss the results and the novelty of the findings. Grammar and style: several typos are present throughout the text.